



# Location controls the findings of ground-based PSC observations

Matthias Tesche[1], Peggy Achtert[2], and Michael C. Pitts[3]

[1]Leipzig Institute for Meteorology (LIM), Leipzig University, Stephanstrasse 3, 04103 Leipzig, Germany
[2]Meteorological Observatory Hohenpeißenberg, German Weather Service (DWD), Germany
[3]NASA Langley Research Center, Hampton, Virginia 23681, USA

**Correspondence:** Matthias Tesche (matthias.tesche@uni-leipzig.de)

**Abstract.** Spaceborne observations of Polar Stratospheric Clouds (PSCs) with the Cloud-Aerosol Lidar with Orthogonal Polarization (CALIOP) aboard the Cloud-Aerosol Lidar and Infrared Pathfinder Satellite Observations (CALIPSO) satellite provide a comprehensive picture of the occurrence of Arctic and Antarctic PSCs as well as their microphysical properties. However, advances in understanding PSC microphysics also require measurements with ground-based instruments, which are often su-
perior to CALIOP in terms of, e.g. time resolution, measured parameters, and signal-to-noise ratio. This advantage is balanced by the location of ground-based PSC observations and their dependence on tropospheric cloudiness. CALIPSO observations during the boreal winters from December 2006 to February 2018 and the austral winters 2012 and 2015 are used to assess the representativeness of ground-based PSC observations with lidar in the Arctic and Antarctic, respectively. Information on tropospheric and stratospheric clouds from the CALIPSO Cloud Profile product (05kmCPro version 4.10) and the Polar Strato-
spheric Cloud (PSC) mask version 2, respectively, is combined on a profile-by-profile basis to identify conditions under which a ground-based lidar is likely to perform useful measurements for the analysis of PSC occurrence. It is found that the location of a ground-based measurement together with the related tropospheric cloudiness can have a profound impact on the derived PSC statistics and that these findings are rarely in agreement with polar-wide results from CALIOP observations. Considering the current polar research infrastructure, it is concluded that the most suitable sites for the expansion of capabilities for ground-
based lidar observations of PSCs are Summit and Villum in the Arctic and Concordia, Troll, and Vostok in the Antarctic.

## 1   Introduction

The existence of Polar Stratospheric Clouds (PSCs) is of critical importance for stratospheric ozone depletion during polar winter. They provide the surface for heterogeneous reactions which transform stable chlorine and bromine species into their highly reactive ozone-destroying states (Lowe and MacKenzie, 2008; Solomon, 1999). PSC formation requires low temperatures that
support the condensation of stratospheric water vapour and nitric acid vapour onto the available stratospheric aerosol particles. These conditions are generally found from December to February in the Arctic and between late May and early October in the Antarctic (Pitts et al., 2018).

Since the early 1990s, airborne and ground-based lidar remote-sensing observations of PSC optical properties have been used to classify PSCs into different types according to their size, shape, and chemical composition (Achtert and Tesche, 2014).
Detailed observations of PSC occurrence and composition are also available from passive remote-sensing observations with



the Michelson Interferometer for Passive Atmospheric Sounding (MIPAS) instrument (Spang et al., 2018). Today, we are confident that PSC particles consist of supercooled liquid ternary solutions (STS), nitric acid trihydrate crystals (NAT), or water ice (ICE); and that PSCs are made up of different mixtures of those three compositions.

Ground-based lidar observations of PSCs are generally performed at the mercy of tropospheric clouds. Since its launch in June 2006, the Cloud-Aerosol Lidar with Orthogonal Polarization (CALIOP) aboard the Cloud-Aerosol Lidar and Infrared Pathfinder Satellite Observations (CALIPSO) satellite (Winker et al., 2009) has been providing a pole-wide view of Arctic and Antarctic PSCs that is unaffected by tropospheric cloudiness. The initial CALIPSO PSC classification scheme employs light-scattering calculations with spherical and non-spherical particles to relate sets of optical parameters to microphysical properties (Pitts et al., 2009, 2013). Recently, the CALIPSO PSC mask version 2 was introduced to correct deficiencies of the initial CALIPSO PSC classification and to improve composition discrimination (Pitts et al., 2018).

Traditionally, two approaches are used to match ground-based lidar measurements to spaceborne observations. Either statistics from a time series of ground-based measurements are compared to those obtained from averaging spaceborne observations for a specific grid-box around the ground site or individual ground-based observations are matched to the data of the closest CALIPSO approach (Snels et al., 2019). Both methods can introduce biases as a result of imperfect temporal or spatial collocation. In addition, ground-based and spaceborne lidar observations of PSCs are often analysed with customised retrieval algorithms that can vary in their definition of different PSC types (Achtert and Tesche, 2014). The combined data set of CALIPSO cloud observations in the troposphere and stratosphere during the Arctic winters from December 2006 to February 2018 and the Antarctic winters 2012 and 2015 presented here allows for an assessment of the representativeness of ground-based lidar measurements of PSCs in a novel way. This paper starts with a description of the data and methods in Section 2. Results are presented and discussed in Section 3 and conclusions are drawn in Section 4.

## 2 Data and methods

### 2.1 Ground sites

Figure 1 and Table 1 provide an overview of the Arctic and Antarctic research stations considered here. The sites were selected because they are accessible, manned year-round, and assumed to provide the necessary infrastructure for ground-based lidar measurements. Sites are also selected to minimise overlap with other research stations. Emphasised are the established PSC observatories Esrange, Sweden (Blum et al., 2005), Eureka, Canada (Donovan et al., 1997), Ny Ålesund, Svalbard (Massoli et al., 2006), and Sodankylä, Finland (Müller et al., 2001) in the Arctic and Belgrano II (Córdoba-Jabonero et al., 2013), Dumont d'Urville (David et al., 1998; Santecesaria et al., 2001), McMurdo (Adriani et al., 2004; Snels et al., 2019), and Syowa (Shibata et al., 2003) in the Antarctic. Also highlighted are stations with a record of lidar measurements that are not specifically dedicated to PSC observations: Alomar (Langenbach et al., 2019), Iqaluit, and Summit (Neely et al., 2013) in the Arctic and Davis in the Antarctic.



## 2.2 Cloud Profile data

Information on tropospheric clouds is taken from the CALIPSO level 2 version 4.10 cloud profile product (05kmCPro.v4.10) which provides information on the vertical extent of different cloud types as well as profiles of the optical properties of clouds with a resolution of 5 km along the CALIPSO ground track and 30-m height bins below 8.2 km height (60-m height bins between 8.2 and 20.2 km height). The extracted parameters are time, latitude, longitude, and the cloud type as provided in the Vertical Feature Mask.

Features that are identified as clouds in the CALIPSO retrieval are further classified into eight cloud types (Liu et al., 2009): (i) low overcast, transparent, (ii) low overcast, opaque, (iii) transition stratocumulus, (iv) low, broken cumulus, (v) altocumulus (transparent), (vi) altostratus (opaque), (vii) cirrus (transparent), and (viii) deep convective (opaque). Ground-based equivalent CALIPSO observations are those that show an absence of tropospheric clouds or only transparent clouds for which a human operator would likely consider performing a ground-based measurement, i.e. transparent altocumulus, cirrus, or a combination of the two. An overview of the number of considered CALIPSO profiles with PSC observations for different tropospheric cloudiness is presented in Table 2.

## 2.3 PSC mask version 2

The CALIOP version 2 PSC detection and composition classification algorithm (CALIPSO PSC mask v2) separates stratospheric cloud features into STS, NAT mixture, ICE, NAT enhanced, and wave ICE. The PSC mask product has an along-track resolution of 5 km, identical to the tropospheric CALIPSO products, and a vertical resolution of 180 m. The new PSC mask corrects known deficiencies in previous versions (Pitts et al., 2009, 2013) and is described in detail in Pitts et al. (2018). A first evaluation with ground-based measurements at Antarctica is presented in Snels et al. (2019).

While all boreal winters from December 2006 to February 2018 are considered in the analysis of Arctic PSCs, only the austral winters of 2012 and 2015 are are included in the analysis of Antarctic PSCs. However, the generally higher occurrence rate of Antarctic PSCs means that a larger number of individual PSC profiles was observed during the two Antarctic winters compared to the 12 considered Arctic winters (see Table 2).

Profiles in the PSC mask v2 product extend down to 8.2 km. They can therefore contain contributions of upper-tropospheric cirrus, as visualised in Figures 13 and 20 of Pitts et al. (2018). To exclude the contribution of such cirrus clouds from our analysis, only height bins above 14.9127 km (down to bin 85) and 13.1140 km (down to bin 96) are considered to represent Arctic and Antarctic PSC, respectively.

## 2.4 Data analysis

Information from the Vertical Feature Mask in the 05kmCPro.v4.10 cloud profile product is used to sum up the number of height bins with different tropospheric cloud types for each CALIPSO profile. This information is used to identify cloud-free conditions (a total of zero counts for each cloud type) and situations with only transparent tropospheric clouds that would still





enable meaningful PSC observations with a ground-based lidar. In addition, all-sky refers to the use of all profiles independent of tropospheric cloudiness.

90 The PSC mask v2 is processed analogous to the Vertical Feature Mask for tropospheric clouds by accumulating the number of height bins with different PSC types for each CALIPSO profile. PSCs that extend over just one height bin are excluded from the analysis. Profiles are referred to as containing a certain a PSC type, for instance STS, if this types was identified in at least one of the PSC height bins. Maps of the occurrence of the accumulated number of height bins related to different PSC types are normalised by the total number of PSC height bins per considered profile or grid box.

95 To enable a combined analysis of cloudiness in the polar troposphere and stratosphere, the data extracted from the 05km-CPro.v4.10 and PSC Mask v2 products are temporally matched and reduced to only those profiles with detected PSCs. The data set is then filtered according to the occurrence of tropospheric clouds and different PSC types. The filtered data is gridded into cells of 1.25° latitude by 2.50° longitude for visualisation of PSC occurrence.

The matched observations of tropospheric and stratospheric clouds allow for a direct comparison of PSC statistics as seen 100 from ground and space independent of the considered instruments. To assess the representativeness of ground-based PSC measurements, PSC statistics are obtained for boxes of 2° latitude by 2° longitude around the sites in Figure 1 and Table 1. True PSC statistics unaffected by tropospheric cloudiness, i.e. during all-sky conditions, can only be obtained with a spaceborne lidar. In contrast, filtering with respect to tropospheric cloudiness is applied to emulate the likely conditions for meaningful ground-based PSC measurements in the CALIPSO data set. Specifically, we assume that a ground-based lidar would only 105 provide meaningful results during conditions with no clouds or only transparent clouds that would not already attenuate the laser beam before it can reach PSC altitudes. This is referred to as the ground-based view. We subsequently separate between observations of (i) a continuously operating ground-based lidar for which all cases of the ground-based view are considered and (ii) a manually operated system for which one third of the cases of the ground-based view was randomly selected. The two ground-based configurations are used to account for sampling effects related to the fact that most ground-based instruments 110 are operated manually and on campaign basis and that the decision to start a measurement, i.e. the assessment of tropospheric cloudiness, is made subjectively by the operator. The purpose is hence to provide an estimate of the potential effects of, e.g. system downtime, logistical problems, and lack of personnel (to list just a few infrastructural challenges in operating a ground-based lidar at a remote location and under harsh conditions) on the inferred PSC statistics.

# 3 Results and discussion

## 3.1 Arctic observations

The normalised PSC occurrence rate in Figure 2a shows that Arctic PSCs are most abundant between 30°W and 90°E and north of 70°N. The pattern of the CALIPSO-derived PSC occurrence rate resembles the MIPAS-based findings in Figure 6b of Spang et al. (2018). Note that Pitts et al. (2018) derived PSC occurrence frequencies for fixed altitudes of $\Theta = 500$ K (around 20 km) and that the PSC area in their Figure 24 is thus smaller than inferred from considering all PSC height levels as done here. 120 Figure 2a also shows that the geography of the Arctic means that most ground-sites are located in areas of relatively low PSC



occurrence. This is levelled by the normalised occurrence rate of suitable conditions for ground-based observations presented in Figure 2b. The region of highest PSC occurrence rate over the north Atlantic coincides with the highest occurrence of opaque tropospheric clouds. While Ny Ålesund could potentially observe the most PSCs in the Arctic, the occurrence rate of good conditions for ground-based lidar measurements is much lower than at the other Arctic sites. In contrast, sites on Greenland

and in the Canadian Arctic show almost no opaque clouds but - with the exception of Villum - also feature a low occurrence rate of PSCs. A similar situation though with a generally lower rate of suitable conditions for ground-based observations is found for Alomar, Esrange, and Sodankylä. However, these sites provide much easier access than the other more remote locations. Tiksi is a site that could potentially provide information on PSCs over the Siberian Arctic.

The occurrence rate of different PSC types in the Arctic for all-sky conditions is shown in Figure 3. The figure reveals that

STS and NAT mixture are most abundant with a region of maximum STS occurrence over the north Atlantic and southern Greenland. The occurrence rates of NAT enhanced and ICE are well below 10% and neither type shows an area of pronounced occurrence. The distribution of wave ICE in Figure 3e shows that this type is restricted regionally to southeastern Greenland, around Iceland, southern Svalbard, the Scandinavian mountain range, and Novaya Zemlya.

Figure 4 provides a local quantification of the Arctic-wide display in Figure 3 for the selected Arctic sites in Table 1 in

the form of the occurrence rate of different PSC types as seen by a spaceborne instrument (all-sky conditions, same as in Figure 3), a continuously operating ground-based instrument (no or only transparent clouds), and a manually operated ground-based instrument (one third of randomly selected CALIPSO profiles in the presence of no or only transparent clouds). For the entire Arctic, the spaceborne view gives a smaller fraction of NAT mixture compared to the ground-based view because the regional minimum in the occurrence rate of NAT mixture (Figure 3b) covers the location of most of the considered ground

sites. This is balanced by a larger fraction of STS for the entire Arctic compared to most ground sites. The occurrence rates of NAT enhanced, ICE, and wave ICE are marginal with a total contribution of less then 10% of all observed PSC height bins. Tropospheric cloudiness would allow for ground-based observations in only about 42% of all Arctic CALIPSO PSC profiles. This causes the slight difference between the three bars related to Arctic-wide observations in Figure 4.

The localised view for 15 ground sites in the Arctic reveals the different impact of tropospheric cloudiness on the statistics

on PSC microphysical properties as expected from Figure 3. Alert and Eureka in the Canadian Arctic and Summit, Thule, and Villum on Greenland, where the conditions for ground-based observations are best (see Figure 2b), show little difference between the spaceborne and the ground-based view. Differences in PSC statistics at those site would more likely be related to the imperfect sampling of a manually operated instrument. The smallest amount of observed CALIPSO PSC profiles is found for Igloolik (183), Iqaluit (249), Myvatn (918), Qeqertarsuaq (848), and Tiksi (326) compared to the other sites where this

number ranges from 2080 for Esrange to 7573 for Ny Ålesund. Consequently, PSC statistics at these sites are much more sensitive to cloudiness and further sub-sampling. A considerable difference between the spaceborne and ground-based view is found in the European Arctic, particularly at Myvatn and Sodankylä. The occurrence rate of STS (ICE) is underestimated (overestimated) at Esrange, Myvatn, and Sodankylä while the opposite is found at Alomar and Ny Ålesund. The ratio of the number of PSC height bins representing the ground-based versus the spaceborne view is given in the third column of Table 1



and allows for the ranking of the ground sites with respect to the occurrence rate of suitable conditions for ground-based measurements.

Apart from the different effect of tropospheric cloudiness, Figure 4 also reveals that statistics of PSC microphysical properties can vary with location. Alert, Eureka, and Thule show STS (NAT mixture) occurrence rates below (above) the Arctic mean of about 30% (60%) while the opposite is the case at Alomar, Esrange, Iqaluit, Myvatn, Ny Ålesund, and Summit where

the occurrence rate of STS exceeds 40% and that of NAT mixture stays below 40%. The highest and lowest occurrence rates of NAT enhanced are found at Igloolik and Alomar, respectively. The other sites show values that are mostly in line with the Arctic mean. ICE is most abundant at Myvatn, Qeqertarsuaq, Sodankylä, and Zackenberg and rarely observed at Alert, Alomar, Eureka, Thule, and Tiksi. Contributions of wave ICE are noticeable only at Myvatn, Sodankylä, and Zackenberg (see Figure 3e) and negligible at the other sites.

## 3.2 Antarctic observations

Figure 5a shows that CALIPSO PSC profiles in the Antarctic are nearly equally distributed around the pole with a higher occurrence rate at higher latitudes. The same is found in the MIPAS climatology (Spang et al., 2018). Tropospheric cloudiness related to conditions that support ground-based lidar measurements Figure 5b is most abundant inland whereas the majority of Antarctic stations is located at the coast to keep logistics manageable. As for the Greenland ice sheet, the elevation of the

better part of Antarctica translates into a complete absence of low-level clouds – the biggest antagonist to atmospheric lidar measurements. Cloudiness is largest upwind from the Antarctic Peninsula. The final column in Table 1 confirms that the lowest occurrence rate of favourable conditions for ground-based lidar measurements of PSCs is found at Marimbio (43%) and San Martín (45%), which are located on the Antarctic Peninsula. The opposite, i.e. an occurrence rate of unity, is true for Concordia and Vostok on the Antarctic Plateau.

The maps of the occurrence rates of different PSC types in the Antarctic during all-sky conditions in Figure 6 show that STS and NAT enhanced are rather homogeneously distributed. A regional minimum in the occurrence of NAT enhanced is found over the West Antarctic Ice Sheet, the Weddell Sea, and parts of Queen Maud Land. This is compensated by higher occurrence rates of ICE. As in the Arctic, wave ICE occurs more locally and is restricted to the Antarctic Peninsula and the border between the Ross Sea and Victoria Land. Despite their layer-based approach on PSC occurrence frequency, Figure 19

in Pitts et al. (2018) presents similar findings regarding the distribution of STS, NAT, and ICE.

The statistics of Antarctic PSC microphysical properties are shown in Figure 7 and vary with location. There are, however, two noticeable differences compared to the situation in the Arctic. Firstly, there is generally little difference in the statistics related to the spaceborne and ground-based view. This is because opaque clouds are less abundant in the Antarctic compared to the Arctic. It is therefore more likely to find reasonable agreement between ground-based an spaceborne PSC observations

at Antarctic sites (Snels et al., 2019) and to observe the same long-term statistics for Antarctic PSCs from ground and space. Secondly, sites such as McMurdo and Vostok show statistics that resemble those obtained for the entire Antarctic. The largest occurrence rates of STS are found at Marimbio, Neumayer III, San Martín, and Troll. However, these values don't exceed those for the entire Antarctic by more than 10 percentage points. The lowest occurrence rate of STS is found at Casey with





a difference of also about 10 percentage points compared to the Antarctic mean. Casey is also the station with the highest
occurrence rate of NAT mixture followed by Mirny. In addition, these two stations show almost no ICE PSCs. The lowest rate
of NAT mixture and the highest rate of ICE (45%-50%) is found at Belgrano II, as this is the only site located in the regional
minimum (maximum) of the occurrence rate of NAT mixture (ICE) revealed in Figure 6. All other sites show ICE occurrence
rates below the Antarctic average. Wave ICE is found only at Jang Bogo (1%) and McMurdo (0.5%).

### 3.3 Location assessment

Figure 8 combines the information on the occurrence rates of PSC and of tropospheric conditions that support PSC observations
with ground-based lidar. This display helps to assess the likelihood for obtaining suitable amounts of data for studying PSCs
from ground-based lidar observations at the sites considered in this study. For the sites to the left of the dashed line that marks
2000 available CALIPSO PSC profiles, the PSC occurrence rate is too low to consider the establishment of a new lidar station
for PSC observations. To the right of the dashed line, further separation is provided by the grey lines that represent different
ratios of cloudiness versus data availability. The most suitable stations for PSC observations can be found to the right of the
1:1 line because they combine a high PSC occurrence rate with a high rate of favourable conditions for PSC observations from
ground. Of the established PSC observatories only Eureka, McMurdo, and Ny Ålesund fall into this category. At Ny Ålesund,
the high occurrence rate of tropospheric clouds is levelled by the also high PSC occurrence rate (see Figure 2). Note that the
assessment in Figure 8 is based entirely on atmospheric conditions and does not consider infrastructural challenges such as
the accessibility, power supply, or availability of facilities at the respective sites; or the training and work load of the stationed
personnel. It is because of this that most of the established PSC observatories fall into a region that could be considered as
less suitable for establishing a ground-site for PSC observations. Nevertheless, the trade-off between PSC occurrence and
tropospheric cloudiness at those sites still creates conditions that allow for meaningful amounts of PSC observations — as
witnessed by the available literature. If new PSC observatories were to be established, the most suitable choices – based solely
on atmospheric conditions – would be Villum, Summit, Zackenberg, Thule, and Alert in the Arctic; and Vostok, Concordia,
Troll, Jang Bogo, Belgrano II, and Neumayer III in the Antarctic.

### 4 Summary and conclusions

There is a rich literature on airborne and ground-based PSC measurements going back to the 1980s. The thus collected time
series have been used to obtain statistics of microphysical properties of PSCs in the Arctic and Antarctic. While the impact
of using different PSC classifications schemes has been assessed in the past (Achtert and Tesche, 2014), there as not yet been
an evaluation of the comparability and the representativeness of the available time series and statistics of ground-based PSC
observations. Here, CALIPSO lidar observations of clouds in the troposphere and stratosphere are used to compare statistics
of PSC microphysical properties as observed (i) from space and ground and (ii) at different ground sites. The data set shows
a strong dependence of PSC microphysical statistics on the location of a ground site in both the Arctic and the Antarctic.



In the Arctic, there is the additional combined effect of the inhomogeneous distribution in the occurrence of both PSCs and tropospheric clouds on the representativeness of ground-based PSC observations with respect to all-sky conditions.

The combination of the occurrence rate of PSCs and of suitable conditions for ground-based PSC observations allows to assess the suitability of a ground site for long-term lidar measurements of PSCs. This suitability is related solely to atmospheric conditions and does not consider challenges with respect to logistics, personnel, or training. According to this definition, mea-
surements at more suitable sites will require less measurement effort to obtain a data set that can be used to infer statistically significant PSC data. This knowledge is important as ground-based lidars are generally more advanced than spaceborne instruments and allow to independently retrieve backscatter and extinction coefficients as well as the particle linear depolarisation ratio at multiple wavelengths and at a better signal-to-noise ratio. Their measurements are therefore invaluable for a better understanding processes related to PSC formation and persistence.

Of the established PSC observatories only Eureka, McMurdo, and Ny Ålesund are found to fall into a category that provides a good balance between PSC occurrence and tropospheric cloudiness. Dumont d'Urville is at the lower end of available PSC observations while Esrange, Sodankylä, and Syowa all show only about 1000 CALIPSO PSC profiles during conditions for ground-based measurements. The occurrence rate of PSCs in the Arctic is much lower than in the Antarctic. Hence, the assessment prevented here is particularly important for Arctic sites. Considering only atmospheric conditions, it is found that
Villum, Summit, Zackenberg, Thule, and Alert would be the best choices for establishing new PSC observatories with state-of-the-art lidar instruments the Arctic. In the Antarctic, this is that case for Vostok, Concordia, Troll, Jang Bogo, Belgrano II, and Neumayer III.

The strong dependence of PSC formation on temperature suggests a crucial role of processes that enhance local cooling (Carslaw et al., 1998; Teitelbaum et al., 2001). These include synoptic or mesoscale events that are generally linked to specific
types of tropospheric cloudiness. It is therefore reasonable to expect a connection between tropospheric cloudiness and the occurrence of PSCs and maybe even different PSC types. Initial studies focussed on individual winters in the Arctic (Achtert et al., 2012) and Antarctic (Wang et al., 2008; Adhikari et al., 2010) show that particularly high and deep-convective cloud systems have a strong effect on PSC formation. This indicates that tropospheric meteorology might be an important driver for the interannual variability in PSC formation and ozone hole recovery. While CALIPSO is operational since 2006, there has not
yet been a thorough assessment of the dependence of the occurrence of different PSC types on tropospheric cloudiness. In the future, the combined CALIPSO data set of clouds in the troposphere and stratosphere presented here will be used to investigate this connection.

*Data availability.*   CALIPSO Cloud Profile data were obtained from the ICARE Data and Services Center (http://www.icare.univ-lille1.fr/). CALIPSO PSC Mask v2 data are available from Michael C. Pitts upon request.



*Author contributions.* MT and PA conceived the study, developed the methodology, and analysed the data. CALIPSO PSC Mask v2 data were provided by MCP. All authors contributed to the discussion of the data and the preparation of the manuscript.

*Competing interests.* The authors declare that they have no conflict of interest.

*Acknowledgements.* We thank the CALIPSO Science team for providing CALIPSO data for tropospheric clouds. This work was supported by the Franco-German Fellowship Programme on Climate, Energy, and Earth System Research (Make Our Planet Great Again – German Research Initiative, MOPGA-GRI) of the German Academic Exchange Service (DAAD), funded by the German Ministry of Education and Research.





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



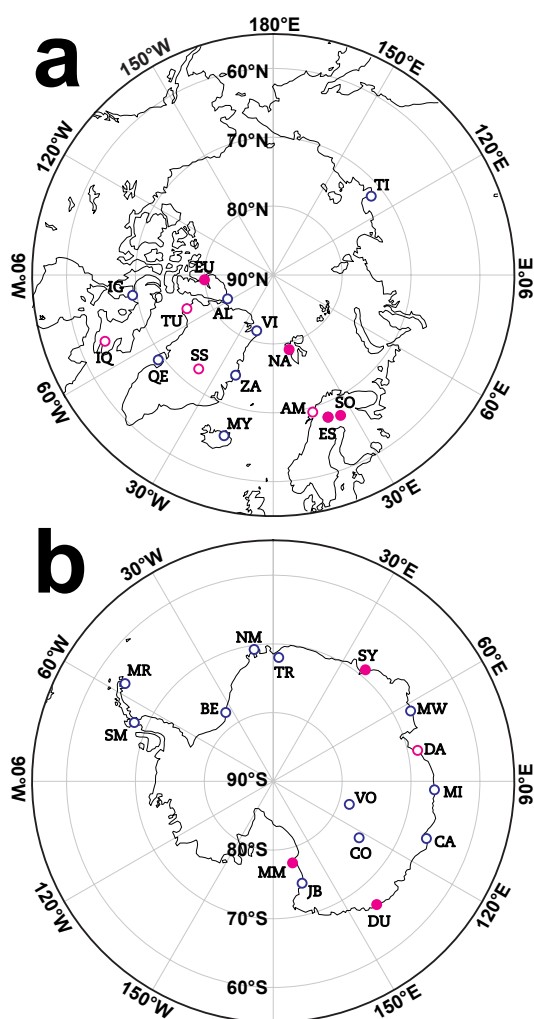

**Figure 1.** Locations of research stations in (a) the Arctic and (b) the Antarctic and their respective abbreviations are listed in Table 1. Red open circles mark stations with atmospheric lidar measurements while red filled circles refer to stations with published PSC measurements. Other stations of potential interest for ground-based PSC observations are marked by blue open circles.



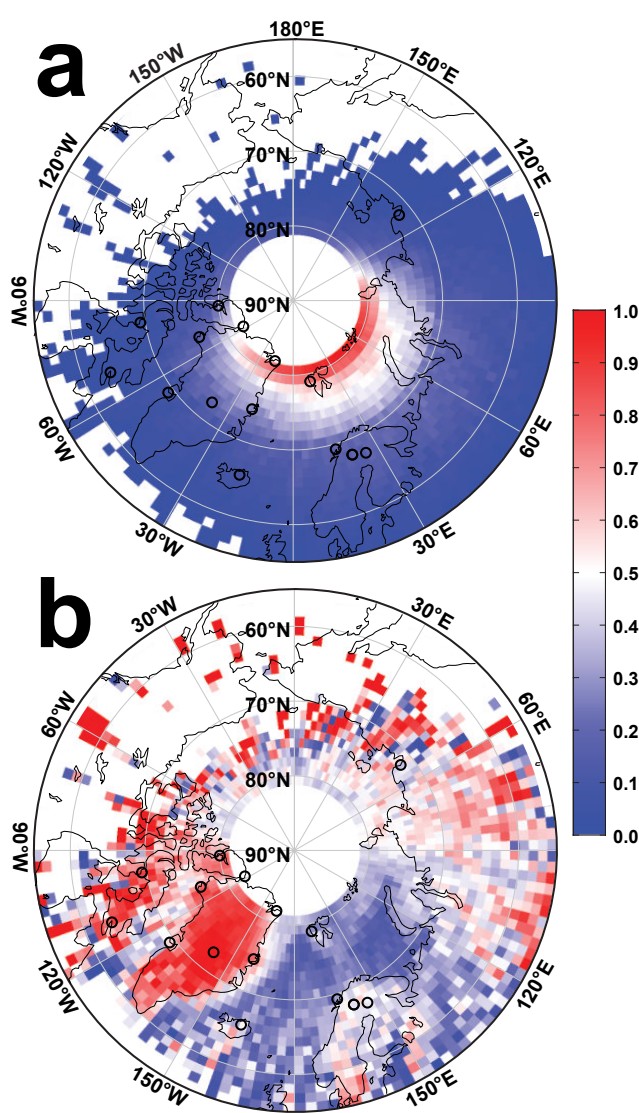

**Figure 2.** Normalised number of CALIPSO profiles with PSCs detected over the Arctic (a, scaled to maximum count of 2478) and (b) the occurrence rate of favourable tropospheric cloud conditions for ground-based lidar measurements (no or only transparent clouds). Black circles mark the locations of lidar ground sites shown in Figure 1 and listed in Table 1.

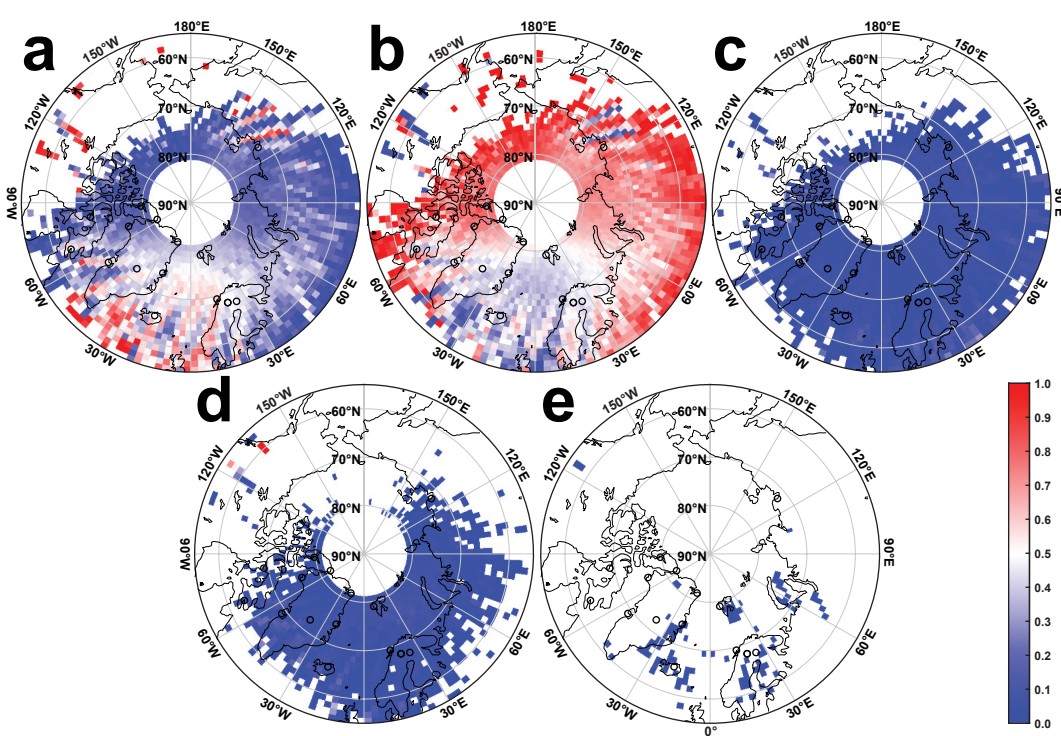

**Figure 3.** Normalised occurrence rate of CALIPSO height bins that contain (a) STS, (b) NAT mixture, (c) NAT enhanced, (d) ICE, and (e) wave ICE for all-sky conditions in the Arctic.



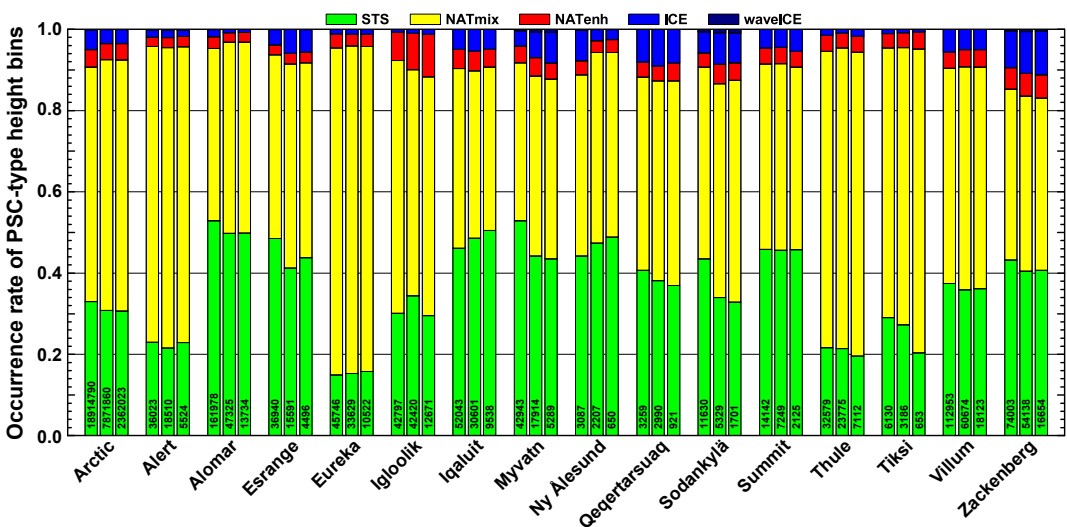

**Figure 4.** Occurrence rate of STS (green), NAT mixtures (yellow), NAT enhanced (red), ICE (blue), and wave ICE (dark blue) for the entire Arctic as well as for the Arctic ground sites listed in Table 1. The three bars per site refer to (i) all-sky conditions in the troposphere (the view of a spaceborne lidar, left), (ii) conditions with no tropospheric clouds or transparent clouds only (the view of a continuously working ground-based lidar, middle), and one third of randomly selected profiles from observations with no tropospheric clouds or transparent clouds only (the view of a manually operated ground-based lidar, right). Numbers refer to the total amount of considered PSC height bins per configuration.





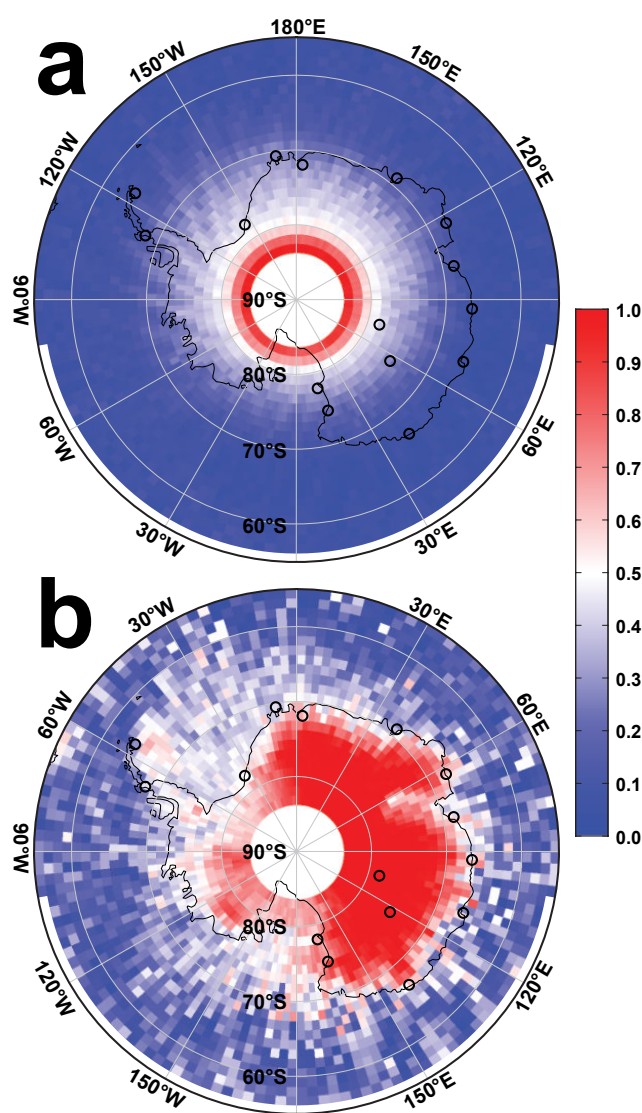

**Figure 5.** Normalised number of CALIPSO profiles with PSCs detected over the Antarctic (a, scaled to maximum count of 2001) and (b) the occurrence rate of favourable tropospheric cloud conditions for ground-based lidar measurements (no or only transparent clouds). Black circles mark the locations of lidar ground sites shown in Figure 1 and listed in Table 1.





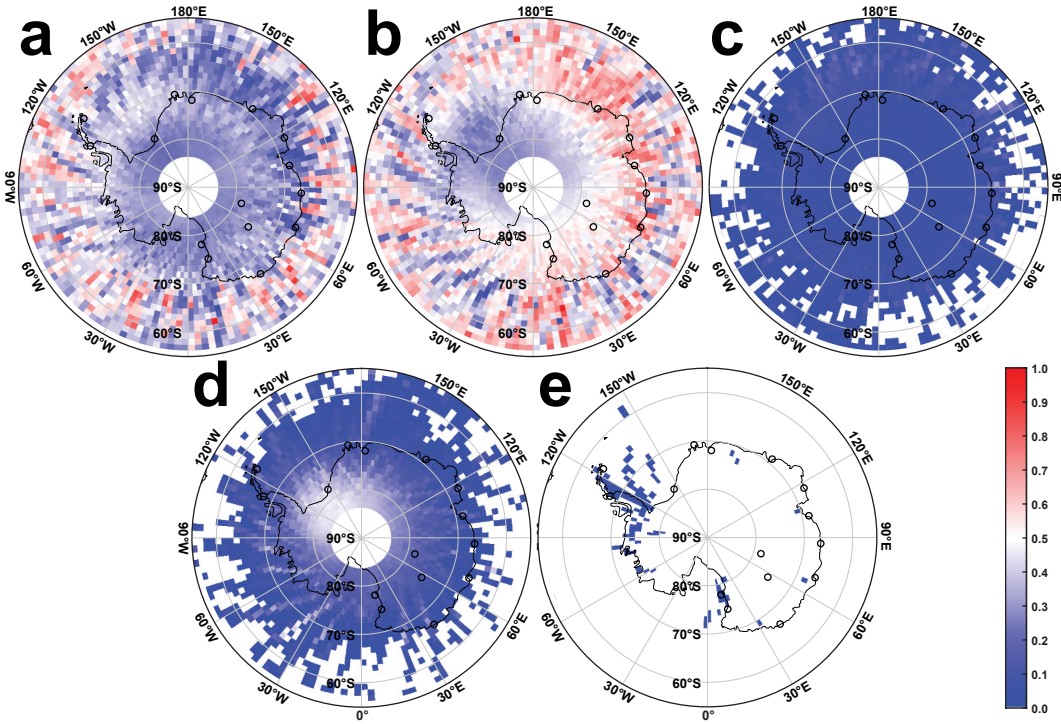

**Figure 6.** Same as Figure 7 but for the Antarctic.

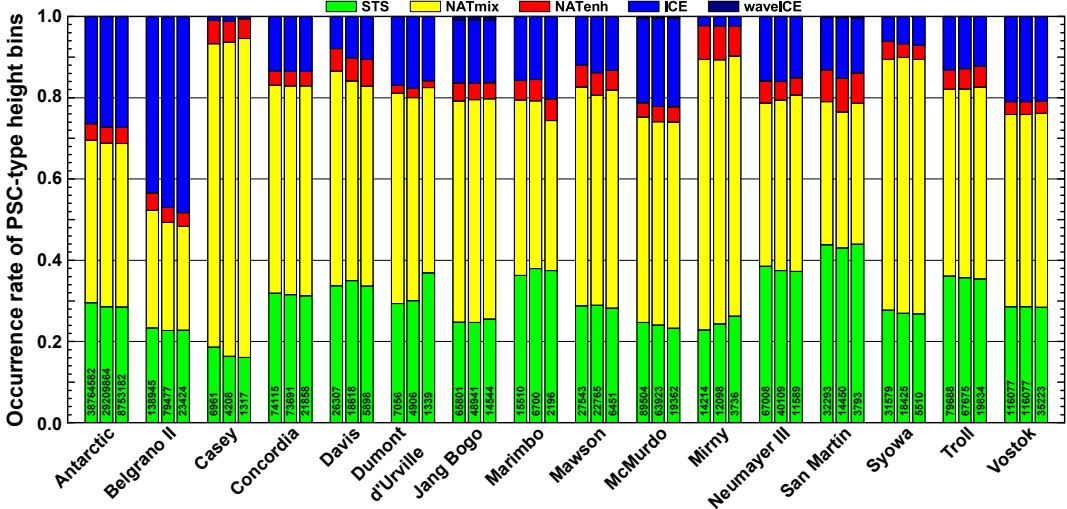

**Figure 7.** Same as Figure 4 but for Antarctic observations.





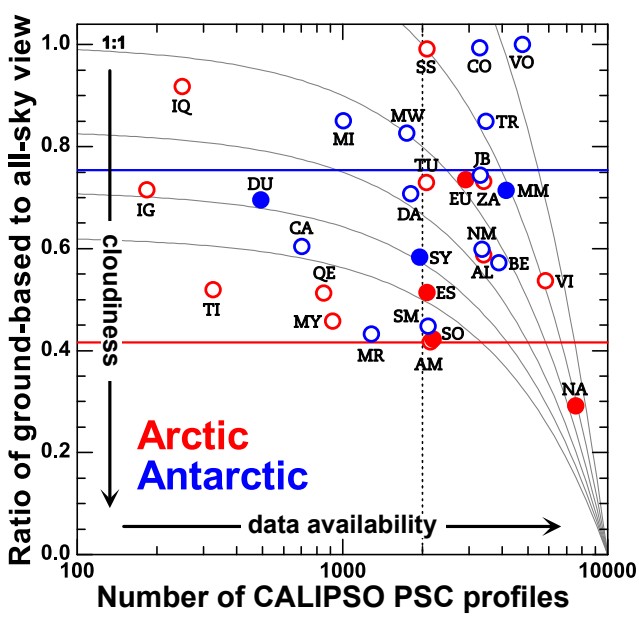

**Figure 8.** Number of CALIPSO PSC profiles in the $4° \times 4°$ grid box centred around the Arctic (red) and Antarctic (blue) ground sites listed in Table 1 versus the ratio of PSC height bins as observed by a ground-based and a spaceborne lidar (columns 3 and 6 in Table 1). Filled symbols mark sites with published PSC climatologies. Horizontal lines mark the values for the entire Arctic and Antarctic, respectively. The vertical dashed line separates stations with more than 2000 CALIPSO PSC profiles from those with fewer observations. Grey lines mark the ratios 0.6:1.0, 0.8:1.0, 1.0:1.0, 1.0:1.2, 1.0:1.4, and 1.0:1.6. Stations abbreviations are given in Table 1.





**Table 1.** Overview of the location of Arctic and Antarctic research stations. Station abbreviations in columns 1 and 5 are used to mark the corresponding sites in Figures 1 and 8. Stations with a deployment of atmospheric lidar instruments are marked with ♣ while those with existing PSC data sets are marked with ♠. R gives the ratio of PSC height bins for tropospheric cloudiness that relates to the data coverage of a ground-based (cloud-free and transparent clouds) and a spaceborne lidar (all-sky).

| | Arctic station | Location | R | | Antarctic station | Location | R |
|---|---|---|---|---|---|---|---|
| AL | Alert, Canada | 82°N, 62°W | 0.59 | BE | Belgrano II, Coats Land ♠ | 78°S, 35°W | 0.57 |
| AM | Alomar, Norway ♣ | 69°N, 16°E | 0.42 | CA | Casey, Vincennes Bay | 66°S, 111°E | 0.60 |
| ES | Esrange, Sweden ♠ | 68°N, 21°E | 0.51 | CO | Concordia, Antarctic Plateau | 75°S, 123°E | 0.99 |
| EU | Eureka, Canada ♠ | 80°N, 86°W | 0.74 | DA | Davis, Princess Elizabeth Land ♣ | 69°S, 78°E | 0.71 |
| IG | Igloolik, Canada | 69°N, 82°W | 0.71 | DU | Dumont d'Urville, Aélie Land ♠ | 66°S, 140°E | 0.70 |
| IQ | Iqaluit, Canada ♣ | 64°N, 69°W | 0.92 | JB | Jang Bogo, Terra Nova Bay | 75°S, 164°E | 0.74 |
| MY | Myvatn, Iceland | 66°N, 17°W | 0.46 | MR | Marambio, Marambio Island | 64°S, 57°W | 0.43 |
| NA | Ny Ålesund, Svalbard ♠ | 79°N, 12°E | 0.29 | MW | Mawson, Mac Robertson Land | 68°S, 63°E | 0.83 |
| QE | Qeqertarsuaq, Greenland | 69°N, 54°W | 0.51 | MM | McMurdo, Ross Island ♠ | 78°S, 167°E | 0.71 |
| SO | Sodankylä, Finland ♠ | 67°N, 27°E | 0.42 | MI | Mirny, Davis Sea | 67°S, 93°E | 0.85 |
| SS | Summit Station, Greenland ♣ | 73°N, 39°W | 0.99 | NM | Neumayer III, Atka Bay | 71°S, 8°W | 0.60 |
| TU | Thule, Greenland ♣ | 77°N, 69°W | 0.73 | SM | San Martín, Barry Island | 68°S, 67°W | 0.45 |
| TI | Tiksi, Russia | 72°N, 129°E | 0.52 | SY | Syowa, Queen Maud Land ♠ | 69°S, 40°E | 0.58 |
| VI | Villum, Greenland | 82°N, 17°W | 0.54 | TR | Troll, Queen Maud Land | 72°S, 3°E | 0.85 |
| ZA | Zackenberg, Greenland | 75°N, 21°W | 0.73 | VO | Vostok, Antarctic Ice Sheet | 78°S, 106°E | 1.00 |

**Table 2.** Number of considered CALIPSO profiles with PSC observations for different tropospheric cloudiness in the Arctic (December 2006 to February 2018) and the Antarctic (winters of 2012 and 2015).

| Tropospheric cloudiness | Arctic | Antarctic |
|---|---|---|
| all-sky | 1000572 | 1676986 |
| cloud-free | 218553 | 402630 |
| transparent | 225600 | 740952 |
| ground-based | 444153 | 1143582 |