# Peer review of "On the best locations for ground-based PSC observations"

_Atmospheric Chemistry and Physics, 2020_

## Referee Comment (RC1) · Anonymous Referee #1 · 12 Oct 2020

Referee report on "Location controls the findings of ground-based PSC observations", authored by Tesche, Achtert and Pitts.

The paper addresses the representativeness of ground-based lidar measurements in the Polar regions with respect to CALIOP (and MIPAS) observations of polar stratospheric clouds. The main conclusion of the paper is the identification of the best sites for PSC observation. To my opinion, the title is not adequately describing the main goal of this work. I would suggest something like "How to find the best locations for ground-based PSC observations", which better expresses the conclusions and recommendations of the authors. The comparison of the two CALIOP datasets (troposphere and PSC v2) and the ground-based lidar observations might produce many interesting results. The paper does not fully explore the potential of this method and also is not

considering possible biases due to the different measurement protocols of CALIOP and ground-based lidars. It would be useful to specify the different categories of ground-based lidars; those measuring in a continuous mode, others "randomly" and still others in a CALIOP-synchronous mode". The authors should also explain that CALIOP is NOT a continuous mode lidar at a certain location, but has overpass frequencies in the order of days at specific local times. This might cause a bias in the statistics. Having at disposition both data sets the authors might also explore the possible correlation between tropospheric cloudiness and PSC occurrence (as they mention in lines 240-247). They also might quantify the bias introduced by prohibitive meteorologic conditions, such as cloud cover in the ground-based dataset, by comparing the PSC occurrence, as observed by CALIOP, with and without cloud cover. I suppose that this could be easily done. An important flaw of the paper is that they apparently are not aware of the fact that a lidar observatory is active at Concordia station since 2014 (see e.g Snels, ACPD 2020 and https://tmf.jpl.nasa.gov/testLidar/NDACC_LWG/sites/dome_c.html).

This is particularly relevant, since the authors recommend Concordia as one of the best sites to perform PSC observations. The authors consider the CALIOP observations as a reference system for the ground-based lidar. When they speak about representativeness they refer to the agreement of the statistics of the ground-based lidar measurements with respect to the CALIOP observations. This is generally speaking an acceptable concept, but there are some caveats. CALIPSO is performing 14-15 orbits per day, which means that the orbits have a separation in longitude of about $180/15 =$ 12 degrees (we have ascending and descending overpasses). At a latitude of 70(80) degrees. 12 degrees of longitude means 450 (225) km of distance between successive overpasses. The authors use boxes of 2 x 2 degrees lat-lon boxes to do their statistics, this means that several days are needed to "fill the boxes". Experience shows that tropospheric clouds and PSCs are not constant over days, often they change during the day. The CALIOP overpasses in a box occur at fixed local times and thus are biased wrt to the random ground-based observations. Synchronized ground-based observations eliminate this bias. If one considers only average statistics, one should take into

account the biases present in the comparison of ground-based lidar observations wrt to CALIOP, due to the different measurement times.

Some stations (McMurdo in the past, Concordia in the present, maybe also Belgrano) synchronize their observations with CALIOP overpasses, and this makes the comparison more reliable. I would suggest that the authors comment on the opportunity to perform synchronized measurements with CALIOP overpasses. The synchronized measurements do not improve the occurrence statistics necessarily, but they make comparison with CALIOP more reliable.

Snels, ACPD, 2020: Snels, M., Colao, F., Shuli, I., Scoccione, A., De Muro, M., Pitts, M., Poole, L., and di Liberto, L.: Quasi-coincident Observations of Polar Stratospheric Clouds by Ground-based Lidar and CALIOP at Concordia (Dome C, Antarctica) from 2014 to 2018, Atmos. Chem. Phys. Discuss., https://doi.org/10.5194/acp-2020-972, in review, 2020. Other comments:

Abstract. Line 8. What do the authors mean by representativeness ? Is it wrt to the CALIOP observations in a lat-lon box or wrt to the overall occurrence statistics in the Northern or Southern Hemisphere ?

Line 12. These findings are rarely in agreement with polar-wide results. . ... Why would one expect an agreement with polar-wide results? Each location is different. It would be more interesting to have an agreement with a "box-region" observed by CALIOP

Line 15. Concordia is already a NDACC lidar observatory since 2014. Data are available on the NDACC web-site.

Line 33 "calculations with" should read "calculations considering. . ."

Line 43: representativeness see comment on line 8

Line 47 : I would prefer "ground stations" instead of "ground sites", "site" already implies "ground"..

Lines 81-83, This line is not very clear for readers that are not familiar with CALIOP data and should be written in a more "reader friendly" way. The 4 digits in the height are not significant and mentioning the bin number is irrelevant.

Line 92 ..if this type. . ...

Line 101 . the 2x2 degrees boxes correspond with 220 x 76 km at 70 degrees of latitude and 220x38 km at 80 degrees latitude. This implies that the box dimensions change with the locations. Does this create a bias on the statistics ?

Line 108: I would add (iii) ground-based observations synchronized with CALIPSO overpasses.

Line 108-113. The authors want to estimate potential biases due to the mode of operation of the ground based stations. The answer is apparently in the small numbers in Figure 7. To my opinion these numbers do not address adequately the question they posed in the introduction, since the difficulties encountered while recording ground-based measurements cannot be simply translated in doing random measurements.(implicating that non random measurements would give different results..). "(ii) a manually operated system for which one third of the cases of the ground-based view was randomly selected." What does this mean and how it works? In most cases the number in the third column is about 1/3 of the second column, except for Tiksi. Why is that? What is the rationale between taking a random 1/3 or just divide by three ?

Line 201. It is not clear what the 1:1 line means, and also the other grey lines like 1.0:1.6 are not clear. The authers write "the grey lines mark the ratios. . ..." But which ratios ?

Line 202 add Concordia Line229 understanding of processes.

Figure 2 . The longitudes in fig b are wrong !

Figure 4 shows the occurrence rate of the different PSC classes as seen by CALIOP, by the ground-based lidar (continuously operating) in clear sky conditions and for manually

operated ground-based stations. This figure is not clear for what concerns the small numbers written in the coloured columns. It would be better to have a Table with these numbers. Then the number of continuously operating lidars is very small.

Figure 5 the longitudes are wrong in fig a

Figure 6 the longitudes are wrong

Table 1. mark Concordia with existing datasets (see NDACC) The authors might indicate in Table 1 (or in a new Table limited to PSC observing stations) which lidars are continuously operated, which are randomly operated (whenever it suits the operator) and which are synchronized with CALIOP overpasses.

---

## Author Comment (AC1) · 13 Oct 2020

We would like to thank the Referee for the valuable input. Please find out point-by-point reply below. Referee comments are given in black, our answers are given in blue.

**Anonymous Referee #1**

The paper addresses the representativeness of ground-based lidar measurements in the Polar regions with respect to CALIOP (and MIPAS) observations of polar stratospheric clouds. The main conclusion of the paper is the identification of the best sites for PSC observation. To my opinion, the title is not adequately describing the main goal of this work. I would suggest something like "How to find the best locations for ground-based PSC observations", which better expresses the conclusions and recommendations of the authors.

We thank the Referee for this suggestion. It is indeed an outcome of the paper to define the best locations for ground-based PSC observations. Nevertheless, we think that the current title is appropriate as we show results for the entire Arctic and Antarctic, respectively, in Figures 2, 3, 5, and 6. These maps provide information that goes further than just the locations of existing research stations.

The comparison of the two CALIOP datasets (troposphere and PSC v2) and the ground-based lidar observations might produce many interesting results. The paper does not fully explore the potential of this method and also is not considering possible biases due to the different measurement protocols of CALIOP and ground-based lidars. It would be useful to specify the different categories of ground-based lidars; those measuring in a continuous mode, others "randomly" and still others in a CALIOP-synchronous mode". The authors should also explain that CALIOP is NOT a continuous mode lidar at a certain location, but has overpass frequencies in the order of days at specific local times. This might cause a bias in the statistics.

We are sorry that our description was not as clear as intended. The scope of this study is to explore the effect of tropospheric cloudiness on what would be observed from ground. The basic assumption is that the CALIPSO lidar can observe all PSCs along its laser beam while tropospheric clouds might attenuate the laser beam before it can reach PSC altitudes. In other words, we look at the same observation from two directions, i.e. from ground and space. In terms of the measurement protocol of a ground station, this implies CALIPSO-synchronous mode. We then separate between scenarios in which measurement at ground are performed (i) during each CALIPSO overpass and (ii) only one third of all CALIPSO overpasses. The first scenario can be realised both with a continuously operating lidar and with a system that is operating during each CALIPSO overpass with downtime in between but only if there is no interference by tropospheric clouds or measurement-inhibiting factors such as maintenance, downtime, or operator availability. The second scenario also refers to CALIPSO-synchronous measurements with the caveat that interfering factors reduce the number of observations to one third. This latter scenario is much more realistic for most polar lidar stations.

We agree that the labelling of a continuously operating lidar and a manually operated lidar was misleading. Accounting for this and the fact that we really only consider the CALIPSO-synchronous scenario, we have dropped the reference to continuously operating and manually operated instruments. We have also revised the text in Section 2.4 to:

"*The matched observations of tropospheric and stratospheric clouds allow for a direct comparison of individual PSC profiles as well as long-term PSC statistics as seen from ground and space independent of the considered instruments. Specifically, the same profile can be evaluated from two perspectives, i.e. from space as well as from the point of view of a ground-based instrument. In that context, the latter perspective translates to a CALIPSO-synchronous measurement protocol at a ground station. True PSC statistics unaffected by tropospheric cloudiness, i.e. during all-sky conditions, at a certain*"

*location can only be obtained with a spaceborne lidar. In contrast, filtering with respect to tropospheric cloudiness is applied to emulate the likely conditions for meaningful ground-based PSC measurements in the CALIPSO data set. Specifically, we assume that a ground-based lidar would only provide meaningful results during conditions with no clouds or only transparent clouds that would not already attenuate the laser beam before it can reach PSC altitudes. This is referred to as the ground-based view of the CALIPSO data set. It provides sampling that is dependent on the CALIPSO return rate and must not be confused with actual ground-based measurements that can provide localised PSC observations in the time range from hours to weeks.*

*We subsequently separate the ground-based view of the CALIPSO data set into two scenarios for which (i) all cases of the ground-based view are considered and (ii) one third of the profiles of the ground-based view was randomly selected. The first scenario corresponds either to a continuously operating lidar or a manually operated system that is active during every single CALIPSO overpass with possible downtime in between without any interference by tropospheric clouds or measurement-inhibiting factors. The second scenario also refers to CALIPSO-synchronous measurements with the caveat that interfering factors reduce the number of measured lidar profiles to one third of what would ideally be possible. This latter scenario is much more realistic as (i) most ground-based lidar instruments are operated manually and on campaign basis, (ii) the decision to start a measurement, i.e. the assessment of tropospheric cloudiness, is made subjectively by the operator, and (iii) infrastructural challenges (e.g. system downtime, logistical problems, and lack of personnel) affect the operation of a ground-based lidar at a remote location and under harsh conditions.*

*To assess the representativeness of ground-based PSC measurements, PSC statistics are obtained for boxes of 2° latitude by 2° longitude around the sites in Figure 1 and Table 1."*

Having at disposition both data sets the authors might also explore the possible correlation between tropospheric cloudiness and PSC occurrence (as they mention in lines 240-247).

The investigation of the connection between tropospheric and stratospheric cloudiness has actually been our motivation from the outset. The present study turned out to be a by-product of this work and we decided to publish it first as makes for a nice stand-alone publication.

They also might quantify the bias introduced by prohibitive meteorological conditions, such as cloud cover in the ground-based dataset, by comparing the PSC occurrence, as observed by CALIOP, with and without cloud cover. I suppose that this could be easily done.

**This is actually the scope of the manuscript.** We use the matched CALIPSO observations of tropospheric and stratospheric cloudiness to show what PSC statistics look like during (i) all-sky conditions (spaceborne view, not possible with ground-based instruments), (ii) situations with tropospheric cloudiness that would still enable PSC observations from ground (view of a ground-based lidar with CALIPSO-synchronous measurement protocol operated during every single CALIPSO overpass), and (iii) situations in which CALIPSO-synchronous operation of a ground-based instrument is affected by cloudiness and other measurement-inhibiting factors. We have revised the description of the data analysis in Section 2.4, the caption of Figure 4, and the discussion of Figures 4 for clarity. We have also made revisions throughout the text to clearly state that the purpose of this work is exactly to quantify the bias introduced by prohibitive meteorological conditions, though we refer to them simply as tropospheric cloudiness.

An important flaw of the paper is that they apparently are not aware of the fact that a lidar observatory is active at Concordia station since 2014 (see e.g Snels, ACPD 2020 and

https://tmf.jpl.nasa.gov/testLidar/NDACC_LWG/sites/dome_c.html). This is particularly relevant, since the authors recommend Concordia as one of the best sites to perform PSC observations.

*We thank the Referee for making us aware of this publication and the measurements at Concordia station. The paper has been added and the Figures and discussion have been revised to account for the existence of PSC measurements at Concordia (see also replies below).*

The authors consider the CALIOP observations as a reference system for the ground-based lidar. When they speak about representativeness they refer to the agreement of the statistics of the ground-based lidar measurements with respect to the CALIOP observations. This is generally speaking an acceptable concept, but there are some caveats. CALIPSO is performing 14-15 orbits per day, which means that the orbits have a separation in longitude of about 180/15 = 12 degrees (we have ascending and descending overpasses). At a latitude of 70(80) degrees. 12 degrees of longitude means 450 (225) km of distance between successive overpasses. The authors use boxes of 2 x 2 degrees lat-lon boxes to do their statistics, this means that several days are needed to "fill the boxes". Experience shows that tropospheric clouds and PSCs are not constant over days, often they change during the day. The CALIOP overpasses in a box occur at fixed local times and thus are biased wrt to the random ground-based observations. Synchronized ground-based observations eliminate this bias. If one considers only average statistics, one should take into account the biases present in the comparison of ground-based lidar observations wrt to CALIOP, due to the different measurement times. Some stations (McMurdo in the past, Concordia in the present, maybe also Belgrano) synchronize their observations with CALIOP overpasses, and this makes the comparison more reliable. I would suggest that the authors comment on the opportunity to perform synchronized measurements with CALIOP overpasses. The synchronized measurements do not improve the occurrence statistics necessarily, but they make comparison with CALIOP more reliable.

*Please see our reply to your other comments regarding the possible measurement protocols at ground stations. We have now clarified that our data set corresponds to a CALIPSO-synchronous measurement protocol at ground stations. We have also dropped the misleading reference to continuously and manually operated instruments at ground and replaced the corresponding statements with more accurate ones.*

Snels, ACPD, 2020: Snels, M., Colao, F., Shuli, I., Scoccione, A., De Muro, M., Pitts, M., Poole, L., and di Liberto, L.: Quasi-coincident Observations of Polar Stratospheric Clouds by Ground-based Lidar and CALIOP at Concordia (Dome C, Antarctica) from 2014 to 2018, Atmos. Chem. Phys. Discuss., https://doi.org/10.5194/acp-2020-972, in review, 2020.

*Thank you. The paper has been added to the list of references.*

**Other comments:**

Abstract. Line 8. What do the authors mean by representativeness ? Is it wrt to the CALIOP observations in a lat-lon box or wrt to the overall occurrence statistics in the Northern or Southern Hemisphere ?

*The term representativeness in our study refers to the statistics derived without and with the effect of tropospheric clouds and other measurement-inhibiting factors that can affect the findings of ground-based lidar instruments. The statement was changed to: "CALIPSO observations during the boreal winters from December 2006 to February 2018 and the austral winters 2012 and 2015 are used to assess the effect of tropospheric cloudiness and other measurement-inhibiting factors on the representativeness of ground-based PSC observations with lidar in the Arctic and Antarctic, respectively."*

Line 12. These findings are rarely in agreement with polar-wide results..... Why would one expect an agreement with polar-wide results? Each location is different. It would be more interesting to have an agreement with a "box-region" observed by CALIOP

One might not expect an agreement with polar-wide results at each site but it is reasonable to assume that some sites are more representative of the larger-scale conditions than others. The agreement in a box region is basically what we do for the selection ground sites.

Line 15. Concordia is already a NDACC lidar observatory since 2014. Data are available on the NDACC web-site.

The Referee is correct. The statement has been changed to "and Mawson, Troll, and Vostok in the Antarctic". We rate Mawson over Jang Bogo due to the proximity of the latter to McMurdo.

Line 33 "calculations with" should read "calculations considering..."

Changed to: "light-scattering calculations that consider spherical and non-spherical particle shapes"

Line 43: representativeness see comment on line 8

Revised as in the reply to comment regarding line 8.

Line 47 : I would prefer "ground stations" instead of "ground sites", "site" already implies "ground"

Ground site has been changed to ground station throughout the text. However, we still use the term site when referring to locations.

Lines 81-83, This line is not very clear for readers that are not familiar with CALIOP data and should be written in a more "reader friendly" way. The 4 digits in the height are not significant and mentioning the bin number is irrelevant.

Thank you for this comment. The section was revised to: "*Because of CALIPSO's top-down viewing geometry, profile start with the uppermost height bin down (bin 1) to the lowermost height bin (bin 583). Profiles in the PSC mask v2 product extend down to 8.2 km. They can therefore contain contributions of upper-tropospheric cirrus, as visualised in Figures 13 and 20 of Pitts et al. (2018). To exclude the contribution of such cirrus clouds from our analysis, only height bins above 14.9 km (smaller than bin 85) and 13.1 km (smaller than bin 96) are considered to represent Arctic and Antarctic PSC, respectively.*"

Line 92 ..if this type....

The statement has been clarified to: "*Profiles are referred to as containing a certain PSC type (i.e. STS, NAT mixture, NAT enhanced, ICE or wave ICE), if this type was identified in at least one of the PSC height bins.*"

Line 101 . the 2x2 degrees boxes correspond with 220 x 76 km at 70 degrees of latitude and 220x38 km at 80 degrees latitude. This implies that the box dimensions change with the locations. Does this create a bias on the statistics ?

We don't think that this has much of an effect on the statistics as the smaller box sizes at higher latitudes are compensated for by the higher CALIPSO return rate at higher latitudes.

Line 108: I would add (iii) ground-based observations synchronized with CALIPSO overpasses.

Please see our earlier replies regarding the reference to ground-based measurements and the revisions of Section 2.4.

Line 108-113. The authors want to estimate potential biases due to the mode of operation of the ground based stations. The answer is apparently in the small numbers in Figure 7. To my opinion these numbers do not address adequately the question they posed in the introduction, since the difficulties encountered while recording ground-based measurements cannot be simply translated in doing random measurements. (implicating that non-random measurements would give different results..). "(ii) a manually operated system for which one third of the cases of the ground-based view was randomly selected." What does this mean and how it works? In most cases the number in the third column is about 1/3 of the second column, except for Tiksi. Why is that? What is the rationale between taking a random 1/3 or just divide by three ?

We are sorry about the confusion. We have revised Section 2.4 and hope that it is now more comprehensible. We do refer the bias to the available data coverage but erroneously assumed this could be synonymous to certain modes of operation. We actually only include what would be CALIPSO-synchronous measurements when we consider a CALIPSO lidar profile from the spaceborne and ground-based perspective. We then screen the data set to find those profiles in which tropospheric cloudiness was unlikely to have attenuated the lidar beam if it was coming from ground (no or only transparent clouds). This corresponds to the optimum data yield for ground-based measurements. However, we know that there is a wide range of factors that reduce the amount of collected data from the optimum data yield. We estimated that even under the worst of circumstances, a ground-based instrument should not provide less than one third of the maximum possible measurements. To get to this sub-set of observations, we randomly selected one third of those CALIPSO profiles that represent what would be observable from ground, i.e. the optimum yield. The statistics were derive subsequently from that subset of profiles. The numbers in Figure 7 refer to the number of PSC height bins in the corresponding category. Because the amount of PSC height bins can vary from profile to profile, the scaling is only about one third and not exactly one third.

Line 201. It is not clear what the 1:1 line means, and also the other grey lines like 1.0:1.6 are not clear. The authors write "the grey lines mark the ratios....." But which ratios ?

We agree that the grey lines in Figure 8 were confusing. We have removed all but one and revised the figure caption to: "*The grey line marks a scale PSC coverage defined as (10000 - x)/10000. Stations to the right of this line show a combination of tropospheric cloudiness and PSC coverage that indicates favourable conditions for ground-based lidar measurements*."

We have also revised the discussion of Figure 8 accordingly.

Line 202 add Concordia

done

Line229 understanding of processes.

of has been added

Figure 4 shows the occurrence rate of the different PSC classes as seen by CALIOP, by the ground-based lidar (continuously operating) in clear sky conditions and for manually operated ground-based stations. This figure is not clear for what concerns the small numbers written in the coloured columns. It would be better to have a Table with these numbers. Then the number of continuously operating lidars is very small.

We state in the figure caption that the numbers refer to the total amount of considered PSC height bins per configuration. The purpose of these numbers is to give an idea about the amount of data

that went into the respective bars. We have increase the size of the figure to improve readability. Please see our previous replies clarifying what was meant with the reference to continuously and manually operated ground-based instruments.

Figure 2 . The longitudes in fig b are wrong! / Figure 5 the longitudes are wrong in fig a / Figure 6 the longitudes are wrong

Thank you for spotting this mistake. It has been corrected.

Table 1. mark Concordia with existing datasets (see NDACC) The authors might indicate in Table 1 (or in a new Table limited to PSC observing stations) which lidars are continuously operated, which are randomly operated (whenever it suits the operator) and which are synchronized with CALIOP overpasses.

A corresponding marker has been added to Concordia in the table. We have also changed the marker style of Concordia in Figure 1b from open blue circle to filled magenta circle and in Figure 8 from open to filled circle to denote that it is a research station with published PSC measurements.

---

## Referee Comment (RC2) · Anonymous Referee #1 · 21 Oct 2020

Referee comment (2)

The answers of the authors to my first comments have been all addressed in a satisfactory way and appropriate corrections have been made. I still have some minor remarks, however.

I still find the title not very descriptive. I think that the title I suggested does not exclude any locations. The best locations can be determined from Figures 3 and 6, without specifying existing stations.

Among the reasons for performing or not performing a measurement from the ground, the authors mention "(ii) the decision to start a measurement, i.e. the assessment of tropospheric cloudiness, is made subjectively by the operator ", While the other two

reasons are "random" with respect to the possibility to observe PSCs, the decision of the operator to perform the measurement in absence of tropospheric clouds is not random, since it already selects a favourable condition.

we randomly selected one third of those CALIPSO profiles that represent what would be observable from ground, i.e. the optimum yield. I don't understand why the authors randomly select one third of useful measurements, taking into account the number of pixels where PSCs are present. It would be sufficient to state that the ground based lidars should be able to perform at least one third of the optimum yield.

In Figures 4 and 7 two kinds of information are mixed. The first is the relative number of possible observations by CALIOP, ground-based lidar and one third of the latter. The second is the relative occurrence rate of the different PSC types at the various stations, as observed by CALIOP (the other columns are derived from CALIOP data). The question is if the small differences of the relative occurrence rates between the three columns is "real" or just "casual".

The caption of Figure 6 should read "Same as Figure 3 but for the Antarctic."

---

## Referee Comment (RC3) · Vincent Noel (Referee) · 29 Oct 2020

In this paper, the authors combine two CALIPSO cloud datasets to evaluate the amount of stratospheric clouds (PSCs) that could be detected by ground-based lidars at various polar locations, taking into account the optical obstruction of the lidar laser beam by tropospheric clouds.

The concept behind this study is simple and smart, relatively straightforward to apply once the datasets are made coincident in time and space, and in this study provide results that will be definitely useful to inform installations of lidar instruments in polar locations. In other words, I think the authors had a very good idea. For the most parts, they executed that idea well: generally the paper is clear and well-written, the figures

convey the important points well, and the conclusions are useful. The article is short, which I appreciate, but perhaps a bit too short. I have a few questions for which I could not find answers in the paper, and I think some of the paper's results could be made clearer (see below).

**Major points**

1. My first major point is that while I think I understand how the authors processed profiles with stratospheric clouds and no tropospheric clouds, I'd like a clarification on how the authors decide, when tropospheric clouds are present, whether these clouds are transparent enough for a ground-based lidar to detect the PSC above (L. 105)? I expect the authors apply a threshold criteria on some integrated property of tropospheric clouds within the profile – is it on the geometrical thickness of the tropospheric clouds, on their optical depth, on something else? The value of the threshold might change from one ground-based lidar to the next, since one lidar with higher SNR might be able to penetrate further than another lidar with a smaller SNR. Also, given a semi-transparent tropospheric cloud with a specific optical depth, a given lidar might be able to detect a relatively bright (larger backscatter) PSC beyond, but not detect a thinner one. Could you comment on how these considerations affect your results, or if they do not affect them at all? Maybe discussing the distribution of opacities of tropospheric clouds the ground-based lidars are supposed to go through would help evaluate if this is an important issue or not. These considerations might lead to location-dependent uncertainties of the approach, according to the distribution of opacities of tropospheric clouds and backscatter of stratospheric clouds over a given location.

2. My second point relates to the presentation of the results by location. Once I understood the premise of the study, the first thing I looked for is a figure presenting the amount of PSCs detectable by a ground-based lidar at each location (taking into account obstruction by tropospheric clouds), relative to the amount of PSCs actually present in the profile (and observable from space). That information might be present in Figure 1 (the numbers in each bar?), or Figure 8 (the y-axis?), but I'm not sure.

Regarding Figure 8, I am not sure I understand it correctly. I am under the impression the authors tried to create a single figure that somehow sums up the potential of each location for ground-based lidar observation of PSCs, but this attempt might be at the cost of ease of interpretation. For instance, the meanings of the grey lines is lost on me. Could you make it clearer somehow if that information is present somewhere in the paper, or add it if it's not there? I understand there is value in having a single figure that ranks locations according to their ground-based performance, but maybe the authors could consider spreading the information it contains on several figures to make it easier to discuss and digest?

3. Another information I'd like to see: given a particular location, if we take the spaceborne-retrieved PSC fraction over a given location as the "truth", how off are the fractions retrieved from the incomplete ground-based retrievals at the same location? This would quantify the error or uncertainty in ground-based PSC retrieval from a given location. Depending on the seasonal variability of PSCs over a given location, it might provide a different way to rank the locations. A location with the best sampling might be affected by a larger error than another with a poorer sampling, if the PSCs over that last location do not change much.

**Minor comments**

1. L.26: "Today, we are confident..." I'm not sure we are that confident. There is definitely a consensus in recent studies that study PSCs to focus on three possible particle types (ICE/STS/NAT), but I'm under the impression this consensus has less to do with actual evidence showing that all PSCs are made of these particle types (meaning in-situ measurements) and more with a standardization around dominant retrieval algorithms and datasets. Please use a less confident statement, or correct my impression with references.

2. L. 77: "only the austral winters of 2012 and 2015 are are included in the analysis of Antarctic PSCs": Why is that? Why not use the same record for both poles? If one

dataset is 3 years long and the other 12 years long, how does it affect our confidence in the results from both poles? (Also "are" is said twice)

3. L. 93: "Maps of the occurrence..." Which maps are we talking about here? If this refers to the upcoming figures, why not wait until the figures are introduced to discuss the maps?

4. L. 92 "a certain a PSC", "this types was"

5. Like another reviewer, I do not think the title is a clear description of what this article is about. Without reading the article it is unclear what the authors did. I understand the authors wanted the title to be more about PSCs and less about location ranking, but I find the current title to be less interesting than what the paper describes. It sounds almost obvious: "Location controls the findings of observations" is always true. The contents of the paper go beyond that, and the title might do the article a disservice. I'm not sure what a better title would be though.

6. The approach presented by the authors here has, in my opinion, applications beyond the polar regions. It could be used to rank the potential of locations to provide ground-based observations of high clouds in other regions (eg Tropics), or evaluate the best use of mobile observation setups during campaigns, etc. Maybe the authors could include a comment to this effect in the conclusion.
* * *

---

## Author Comment (AC2) · 2 Nov 2020

We would like to thank the Referee for the valuable input. Please find out point-by-point reply below. Referee comments are given in black, our answers are given in blue.

**Anonymous Referee #1, follow-up comment**

The answers of the authors to my first comments have been all addressed in a satisfactory way and appropriate corrections have been made. I still have some minor remarks, however.

Thank you for the positive feedback.

I still find the title not very descriptive. I think that the title I suggested does not exclude any locations. The best locations can be determined from Figures 3 and 6, without specifying existing stations.

We have thought about the title suggested by the Referee (*How to find the best locations for ground-based PSC observations*) and propose to change our original title (*Location controls the findings of ground-based PSC observations*) to **On the best locations for ground-based PSC observations.**

Among the reasons for performing or not performing a measurement from the ground, the authors mention "(ii) the decision to start a measurement, i.e. the assessment of tropospheric cloudiness, is made subjectively by the operator ", While the other two reasons are "random" with respect to the possibility to observe PSCs, the decision of the operator to perform the measurement in absence of tropospheric clouds is not random, since it already selects a favourable condition.

We agree with the Referee that point (ii) is not as random as the other two points, as an operator is generally capable of identifying cloud-free conditions. What we are referring to here, however, is related to our own experience on deciding whether or not to start a measurement with a manually operated instrument in the presence of clouds. In particular, a measurement could be started in the presence of tropospheric clouds that inhibit PSC observations. An operator might decide to stop the measurement if this cloud deck does not dissolve as expected and the clouds might dissolve after the end of the measurement.

"we randomly selected one third of those CALIPSO profiles that represent what would be observable from ground, i.e. the optimum yield". I don't understand why the authors randomly select one third of useful measurements, taking into account the number of pixels where PSCs are present. It would be sufficient to state that the ground based lidars should be able to perform at least one third of the optimum yield.

The rational for picking the factor of one-third is indeed that we assume that a ground based lidars should be able to perform at least one third of the optimum yield. The intention of sub-sampling the CALIPSO-observations related to the optimum yield, however, is to asses if random sampling of the optimum yield, i.e. subsampling of the dataset in an effort to account for the inhibiting factors imposed on a real-world ground station, would lead to any changes in the overall statistics on PSC type occurrence. Figures 4 and 7 show that this could be the case at some ground stations.

In Figures 4 and 7 two kinds of information are mixed. The first is the relative number of possible observations by CALIOP, ground-based lidar and one third of the latter. The second is the relative occurrence rate of the different PSC types at the various stations, as observed by CALIOP (the other columns are derived from CALIOP data). The question is if the small differences of the relative occurrence rates between the three columns is "real" or just "casual".

The figures provide the occurrence rate of different PSC types related to the three considered conditions viewpoints (all cases of all-sky conditions, all cases of transparent or no clouds, and one third of all cases of transparent or no clouds). The numbers that refer to the amount of considered

PSC height bins provide complementary information that allows to assess the representativity of the measurements, i.e. the statistics become less trustworthy is the number of considered cases falls below a certain level. Our best assessment of the Referee's question is that differences between the first column (all-sky conditions) and the other two columns are real as they describe the effect of tropospheric cloudiness on the obtained statistics. We already state in the discussion of Figure 4: "*The localised view for 15 ground stations in the Arctic reveals the impact of tropospheric cloudiness on the statistics on PSC microphysical properties as expected from Figure 3.*"

In an ideal world, there should be no difference between the second and third column and differences should be casual. Nevertheless, there are stations with a considerable difference in those columns. For such stations (e.g. Igloolik or Tiksi) the sub-sampled data set becomes too small to conclude that the differences are real.

We also realised that the numbers in Figure 4 were mixed up for the different stations. This has now been corrected.

The caption of Figure 6 should read "Same as Figure 3 but for the Antarctic."

Correct. Changed as suggested.

---

## Author Comment (AC3) · 2 Nov 2020

We would like to thank Vincent Noel for the valuable input. Please find out point-by-point reply below. Referee comments are given in black, our answers are given in blue.

**Vincent Noel (Referee #2)**

In this paper, the authors combine two CALIPSO cloud datasets to evaluate the amount of stratospheric clouds (PSCs) that could be detected by ground-based lidars at various polar locations, taking into account the optical obstruction of the lidar laser beam by tropospheric clouds.

The concept behind this study is simple and smart, relatively straightforward to apply once the datasets are made coincident in time and space, and in this study provide results that will be definitely useful to inform installations of lidar instruments in polar locations. In other words, I think the authors had a very good idea. For the most parts, they executed that idea well: generally the paper is clear and well-written, the figures convey the important points well, and the conclusions are useful. The article is short, which I appreciate, but perhaps a bit too short. I have a few questions for which I could not find answers in the paper, and I think some of the paper's results could be made clearer (see below).

Thank you for the overall positive feedback. Please find our detailed replies below.

**## Major points**

1. My first major point is that while I think I understand how the authors processed profiles with stratospheric clouds and no tropospheric clouds, I'd like a clarification on how the authors decide, when tropospheric clouds are present, whether these clouds are transparent enough for a ground-based lidar to detect the PSC above (L. 105)? I expect the authors apply a threshold criteria on some integrated property of tropospheric clouds within the profile – is it on the geometrical thickness of the tropospheric clouds, on their optical depth, on something else? The value of the threshold might change from one ground-based lidar to the next, since one lidar with higher SNR might be able to penetrate further than another lidar with a smaller SNR.

We are sorry that this important point was not clear. Our approach is actually much simpler and doesn't require the use of threshold values or any information on cloud geometrical and optical thickness. For every matched profiles of tropospheric and stratospheric cloud observations, we check the cloud types in the 05kmCPro Vertical Feature Mask. We consider a profile as representing conditions under which a ground-based measurement could be performed, if the Vertical Feature Mask (i) shows no tropospheric clouds at all, (ii) shows only altocumulus (transparent), i.e. cloud type (v) in Section 2.2, (iii) shows only cirrus (transparent), i.e. cloud type (vii) in Section 2.2, or (iv) shows both altocumulus (transparent) and cirrus (transparent). As soon as any other type of tropospheric clouds in present in a profile (any of the four low-level cloud types, altocumulus (opaque), or deep convective (opaque), see Section 2.2), we consider this profile to represent conditions that are unsuitable for a ground-based measurement. Our definition of transparent clouds is already given in Section 2.2. For clarity, we have revised the first paragraph in Section 2.4 to:

*"Information on cloud type from the Vertical Feature Mask in the 05kmCPro.v4.10 cloud profile product is used to sum up the number of height bins with different tropospheric cloudiness for each CALIPSO profile. This information is used to identify cloud-free conditions (a total of zero counts for each of the eight cloud types) and situations with only transparent tropospheric clouds that would still enable meaningful PSC observations with a ground-based lidar, i.e. altocumulus (transparent), cirrus (transparent), or a combination of the two. In addition, all-sky refers to the use of all profiles independent of tropospheric cloudiness."*

Also, given a semi-transparent tropospheric cloud with a specific optical depth, a given lidar might be able to detect a relatively bright (larger backscatter) PSC beyond, but not detect a thinner one. Could you comment on how these considerations affect your results, or if they do not affect them at all? Maybe discussing the distribution of opacities of tropospheric clouds the ground-based lidars are supposed to go through would help evaluate if this is an important issue or not. These considerations might lead to location-dependent uncertainties of the approach, according to the distribution of opacities of tropospheric clouds and backscatter of stratospheric clouds over a given location.

These considerations have no effect on our results as we don't consider geometrical thickness of opacity of the tropospheric clouds. Instead, we rely on the CALIPSO cloud typing which depends on feature altitude (cloud top height) and opacity (whether or not clear sky can be detected below a feature). Because the CALIPSO laser emits less power than most ground-based lidar instruments for PSC observations, we are confident that a cloud that is transparent in a CALIPSO measurement would also be transparent in a ground-based observation.

2. My second point relates to the presentation of the results by location. Once I understood the premise of the study, the first thing I looked for is a figure presenting the amount of PSCs detectable by a ground-based lidar at each location (taking into account obstruction by tropospheric clouds), relative to the amount of PSCs actually present in the profile (and observable from space). That information might be present in Figure 1 (the numbers in each bar?), or Figure 8 (the y-axis?), but I'm not sure.

This is indeed the central information we want to convey by this work. We are sorry to hear that it was hard to figure out the actual numbers. The information on the fraction of PSCs that are observable with a ground-based lidar at a certain location can be taken from (i) the maps in Figures 2b and 5b (occurrence rate of favourable tropospheric cloud conditions for ground-based lidar measurement), (ii) the ratio of the numbers in Figures 4 and 7 (number of PSC height bins during conditions with no tropospheric clouds or transparent clouds only (middle bar) divided by number of PSC height bins during all-sky conditions (left bar)), and (iii) the y-axis in Figure 8 (ratio of ground-based to all-sky view, this was calculated following (ii)).

We have revised the text throughout the manuscript so that the information can be extracted more straightforwardly.

Regarding Figure 8, I am not sure I understand it correctly. I am under the impression the authors tried to create a single figure that somehow sums up the potential of each location for ground-based lidar observation of PSCs, but this attempt might be at the cost of ease of interpretation. For instance, the meanings of the grey lines is lost on me. Could you make it clearer somehow if that information is present somewhere in the paper, or add it if it's not there? I understand there is value in having a single figure that ranks locations according to their ground-based performance, but maybe the authors could consider spreading the information it contains on several figures to make it easier to discuss and digest?

We are sorry for the confusion regarding Figure 8. We agree that the interpretation of this figure was not straightforward. Following the suggestion of the other referee, we have already removed all but one of the grey lines and revised the figure caption to: "*The grey line marks a scale PSC coverage defined as (10000 - x)/10000. Stations to the right of this line show a combination of tropospheric cloudiness and PSC coverage that indicates favourable conditions for ground-based lidar measurements.*"

We have revised the discussion of Figure 8 accordingly. We chose the display in Figure 8 as it nicely presents the two factors that define the rate of success for PSC measurements at a certain ground

station: (i) the effect of tropospheric cloudiness (How often can we measure up to the stratosphere (while PSCs are present)?) and (ii) the occurrence rate of CALIPSO profiles that contain PSCs (How often will there be PSCs?). All stations to the right of the grey line are those that we consider to perform particularly well. This shows for instance that at Ny Alesund, the high PSC occurrence rate compensates for the low occurrence rate of favourable conditions for ground-based PSC measurements – leading to an overall favourable station location.

The information in Figure 8 can be used to produce a simple ranking of stations by multiplying the x and y values. The stations listed in the Abstract and the third paragraph of the Summary are based on such a ranking.

3. Another information I'd like to see: given a particular location, if we take the spaceborne-retrieved PSC fraction over a given location as the "truth", how off are the fractions retrieved from the incomplete ground-based retrievals at the same location? This would quantify the error or uncertainty in ground-based PSC retrieval from a given location. Depending on the seasonal variability of PSCs over a given location, it might provide a different way to rank the locations. A location with the best sampling might be affected by a larger error than another with a poorer sampling, if the PSCs over that last location do not change much.

We might have misunderstood the Referee's comment but the outcome of the PSC classification at different sites for different conditions of tropospheric cloudiness is exactly what is shown in Figures 4 and 7. These figures show the occurrence frequency of different PSC constituents for all-sky conditions (the "true" values), for favourable conditions for ground-based lidar measurements (the ground-based instrument measures whenever tropospheric clouds allow), and for conditions where external circumstances allow for only one third of the optimally possible measurements. We find different effects of tropospheric cloudiness. We also see that locations with poorer sampling tend to show a larger difference between the spaceborne and ground-based view. However, we are only looking at the long-term distribution of PSCs with different composition here and did not consider any seasonal variation.

In any case we would like to ask the Referee to confirm that this is what was meant by the comment.

**Minor comments**

1. L.26: "Today, we are confident..." I'm not sure we are that confident. There is definitely a consensus in recent studies that study PSCs to focus on three possible particle types (ICE/STS/NAT), but I'm under the impression this consensus has less to do with actual evidence showing that all PSCs are made of these particle types (meaning in-situ measurements) and more with a standardization around dominant retrieval algorithms and datasets. Please use a less confident statement, or correct my impression with references.

Thank you for pointing out that the availability of PSC in-situ measurements is still low. We have mitigated the statement to: *"Today, there is consensus that…"*

2. L. 77: "only the austral winters of 2012 and 2015 are are included in the analysis of Antarctic PSCs": Why is that? Why not use the same record for both poles? If one dataset is 3 years long and the other 12 years long, how does it affect our confidence in the results from both poles? (Also "are" is said twice)

This is a fair comment. We started looking at the coincidence of PSCs and tropospheric clouds in the Arctic based on the full data set for this pole. We later realised that a comprehensive documentation of the method and results in a research publication should consider both poles and this is what we did. However, the much larger amount of CALIPSO PSC observations translates into an increased

amount of available data which is further doubled because we consider CALIPSO profiles for both tropospheric and stratospheric clouds, i.e. APro and PSCMask files. We started with two years of Antarctic measurements and found that those actually include more CALIPSO PSC profiles than the entire Arctic data set. So on the one hand, the volume of data is comparable at both poles. On the other hand, we checked that the Antarctic observations are in line with *Pitts et al.* (2018, https://doi.org/10.5194/acp-18-10881-2018). This means that we are confident that including a longer time series of Antarctic observations does not affect the overall conclusions regarding the assessment the representativity of long-term lidar measurements from ground.

Also, we have deleted the second are.

3. L. 93: "Maps of the occurrence..." Which maps are we talking about here? If this refers to the upcoming figures, why not wait until the figures are introduced to discuss the maps?

The normalisation is part of the data analysis methodology which is why we present it in Section 2.4. However, we have moved the statement to the next paragraph after data gridding is mentioned. In addition, we have added a reference to the figures for which the normalisation has been applied to:

*"Maps of the occurrence of the accumulated number of height bins related to different PSC composition are normalised by the total number of PSC height bins per considered grid box (see Figures 2, 3, 5, and 6)."*

4. L. 92 "a certain a PSC", "this types was"

The statement has been revised to:

*"A CALIPSO profiles is referred to as containing a certain a PSC composition (e.g. STS-containing or ICE-containing) if the respective component is identified in at least one of the PSC height bins."*

We have also replaced the reference to PSC type with PSC composition after the introduction.

5. Like another reviewer, I do not think the title is a clear description of what this article is about. Without reading the article it is unclear what the authors did. I understand the authors wanted the title to be more about PSCs and less about location ranking, but I find the current title to be less interesting than what the paper describes. It sounds almost obvious: "Location controls the findings of observations" is always true. The contents of the paper go beyond that, and the title might do the article a disservice. I'm not sure what a better title would be though.

Following the concern of both reviewers, we have revised the title to: ***"On the best locations for ground-based PSC observation."***

6. The approach presented by the authors here has, in my opinion, applications beyond the polar regions. It could be used to rank the potential of locations to provide ground-based observations of high clouds in other regions (eg Tropics), or evaluate the best use of mobile observation setups during campaigns, etc. Maybe the authors could include a comment to this effect in the conclusion.

The Referee is correct. The methodology can be adapted to find suitable locations for observations of mid-level or high clouds or elevated aerosol layers at which the effect of measurement-inhibiting low clouds is minimal. A corresponding statement has been added to the Conclusions:

*"In addition, the methodology presented here can be easily adapted to assess the effect of low-level clouds on tropospheric observations. For instance, it can be used to find locations for measurement campaigns or long-term observatories at which the measurement-inhibiting effect of opaque clouds has a minimum impact on the observational cover of mid-level or high clouds and elevated tropospheric and stratospheric aerosol layers."*

---

## Referee Comment (RC4) · Vincent Noel (Referee) · 3 Nov 2020

I am satisfied with most of the answers to my original comments.

The comment in which I request for a result was not clear. I'll try to do better below.

When PSC measurements are available from a ground-based site, I would expect the first result (before PSC speciation) presented to be the PSC Fraction, which would be defined (by analogy with tropospheric clouds) as the ratio of the number of lidar profiles in which a PSC can be detected, divided by the number of lidar profiles that sample the stratosphere over that location. 100% would mean that all sampled profiles contain a PSC, 50% half of the sampled profiles contain a PSC, etc. This number would inform on the ubiquity of PSCs over the considered area.

[Figure]

Using the authors' methodology, it should be possible to document, over a given location, the actual PSC Fraction (by considering all the profiles sampled by CALIPSO over that area), and the PSC Fraction that would be retrieved from a ground-based lidar (by considering only the profiles that would see the stratosphere considering the presence of opaque tropospheric clouds). From these results one could document the error in retrieved PSC Fraction over all the considered locations. That error might provide an additional data point to rank locations, as locations with smallest errors would enable the most accurate representation of PSC frequency. The numbers retrieved in this fashion would probably align with the accuracy of PSC speciation by location.

---

## Author Comment (AC4) · 4 Nov 2020

Please find our reply below. Referee comments are given in black, our answers are given in blue.

**Vincent Noel (Referee #2)**

I am satisfied with most of the answers to my original comments.

The comment in which I request for a result was not clear. I'll try to do better below.

When PSC measurements are available from a ground-based site, I would expect the first result (before PSC speciation) presented to be the PSC Fraction, which would be defined (by analogy with tropospheric clouds) as the ratio of the number of lidar profiles in which a PSC can be detected, divided by the number of lidar profiles that sample the stratosphere over that location. 100% would mean that all sampled profiles contain a PSC, 50% half of the sampled profiles contain a PSC, etc. This number would inform on the ubiquity of PSCs over the considered area.

Using the authors' methodology, it should be possible to document, over a given location, the actual PSC Fraction (by considering all the profiles sampled by CALIPSO over that area), and the PSC Fraction that would be retrieved from a ground-based lidar (by considering only the profiles that would see the stratosphere considering the presence of opaque tropospheric clouds). From these results one could document the error in retrieved PSC Fraction over all the considered locations. That error might provide an additional data point to rank locations, as locations with smallest errors would enable the most accurate representation of PSC frequency. The numbers retrieved in this fashion would probably align with the accuracy of PSC speciation by location.

We would like to thank Vincent Noel for the follow-up comment. We now understand what the Referee was looking for. Right now, our statistics are restricted to those CALIPSO profiles for which PSCs have been detected. The Referee would like to see what the findings would look like if we were to normalise by the total number of CALIPSO profiles rather than only the number of CALIPSO profiles that show PSCs. Such plots have now been added to Figures 2 and 5. Figures 2b and 5b now show the ratio of all CALIPSO PSC profiles versus all CALIPSO profiles (i.e. the PSC occurrence rate) while Figures 2d and 5d show the ratio of PSC profiles with suitable tropospheric cloudiness for ground-based lidar measurements versus all CALIPSO profiles.

We understand the rationale of the Referee's question regarding the effect of PSC occurrence rate. However, our own experience with running a manually operated ground-based lidar for PSC observations shows that PSC occurrence rate is an ambiguous measure. First, it can only be defined properly in terms of a reference number for normalisation if a ground-based instrument is run continuously or according to a schedule with fixed measurement times. This is often complicated for a manually operated system run by a small team as measurement times are adapted to PSC occurrence. Second, measurements might not be performed if PSCs are absent to save laser lifetime, to perform calibration measurements, or to simply give the operator some time to rest. Finally, PSC statistics are generally obtained only for those lidar profiles that show PSCs.

Nevertheless, we appreciate the Referees suggestion and have revised Figure 2 and 5 accordingly so that the readers can also get an impression of PSC occurrence rates from the polar-wide plots. In addition, we have added the effect of PSC coverage for additional guidance towards finding the best location for ground-based PSC measurements as colour coding to Figure 8. This addition complements the current discussion of Figure 8 but doesn't change the conclusion of the analysis.

Figures 2 and 5 now look like this:

[revised manuscript text omitted]

---

## Author Comment (AC5) · 26 Nov 2020

The comment was uploaded in the form of a supplement:
https://acp.copernicus.org/preprints/acp-2020-930/acp-2020-930-AC5-supplement.pdf